# Clinician and patient perspectives on the barriers and facilitators to physical rehabilitation in intensive care: a qualitative interview study

Huw R Woodbridge [ORCID] ,[1,2] Christine Norton,[3] Mandy Jones,[4] Stephen J Brett [ORCID] ,[1,2] Caroline M Alexander,[1,2] Anthony C Gordon[1,2]

[1]Imperial College Healthcare NHS Trust, London, UK
[2]Department of Surgery and Cancer, Imperial College London, London, UK
[3]Florence Nightingale Faculty of Nursing, Midwifery and Palliative Care, King's College London, London, UK
[4]Brunel University London, London, UK

**Correspondence to**
Dr Huw R Woodbridge;
h.woodbridge@imperial.ac.uk

## ABSTRACT

**Objectives** The objective of this study is to explore patient, relative/carer and clinician perceptions of barriers to early physical rehabilitation in intensive care units (ICUs) within an associated group of hospitals in the UK and how they can be overcome.

**Design** Qualitative study using semi-structured interviews and thematic framework analysis.

**Setting** Four ICUs over three hospital sites in London, UK.

**Participants** Former ICU patients or their relatives/carers with personal experience of ICU rehabilitation. ICU clinicians, including doctors, nurses, physiotherapists and occupational therapists, involved in the delivery of physical rehabilitation or decisions over its initiation.

**Primary and secondary outcomes measures** Views and experiences on the barriers and facilitators to ICU physical rehabilitation.

**Results** Interviews were carried out with 11 former patients, 3 family members and 16 clinicians. The themes generated related to: safety and physiological concerns, patient participation and engagement, clinician experience and knowledge, teamwork, equipment and environment and risks and benefits of rehabilitation in intensive care. The overarching theme for overcoming barriers was a change in working model from ICU clinicians having separate responsibilities (a multidisciplinary approach) to one where all parties have a shared aim of providing patient-centred ICU physical rehabilitation (an interdisciplinary approach).

**Conclusions** The results have revealed barriers that can be modified to improve rehabilitation delivery in an ICU. Interdisciplinary working could overcome many of these barriers to optimise recovery from critical illness.

## INTRODUCTION

The importance of physical rehabilitation of critically ill patients has been recognised because of the prevalence of acute muscle weakness and wasting,[1–3] and longer-term substantial physical disability measured in this patient group.[2 4 5] Physical rehabilitation consists of physical activity interventions (typically mobilising in or away from the bed) that are begun once a patient has reached

### STRENGTHS AND LIMITATIONS OF THIS STUDY

⇒ This study explored a range of perspectives on the barriers to intensive care unit (ICU) rehabilitation (including clinicians and services users), thus eliciting in depth information to reveal a breadth of experiences of barriers and facilitators.

⇒ Thematic framework analysis was used, which enables a systematic approach to organising data, facilitating in-depth exploration of the range of views within themes and between participant groups.

⇒ Patient and family recall of their experiences may have been impacted by the time from intensive care admission to interview; however, interviews took place at the first follow-up opportunity to minimise this effect.

⇒ Efforts were made to gain a range of perspectives using purposive sampling; however, fewer family members or carers took part in this study than former ICU patients.

physiological stability.[6–8] Beginning physical rehabilitation at an appropriate dose while patients are still in an intensive care unit (ICU) can improve physical function while in hospital and expedite discharge,[9] although implementing rehabilitation at a higher dose is not necessarily beneficial.[10] However, when measured, there is a concern that the actual amount of formal physical rehabilitation delivered and patient participation in exercise while in intensive care are low.[11–17]

Studies have previously measured the barriers to implementing rehabilitation, the majority of which use a quantitative approach.[18] However, a qualitative approach is better-suited to exploring interpersonal relations[19] and, therefore, potential barriers relating to team working and patient interactions. Where a qualitative approach has been used, issues of communication and differences in opinion between clinicians[20–25] and difficulty in providing rehabilitation in an

environment where demands on staff and patient time change quickly have been highlighted.[23 26 27] However, the lack of rehabilitation in intensive care continues despite this current understanding of the barriers. Importantly, there is a lack of in-depth knowledge of barriers in a UK setting, which includes views of multiple stakeholders such as ICU clinicians from different professions involved in implementing rehabilitation, as well as patients and family members.[22 28–30]

The objective of this study, therefore, was to explore service users (patients or their relatives/carers) and clinician perceptions of barriers to early physical rehabilitation in ICUs within an associated group of hospitals in the UK and how they can be overcome.

## METHODS

### Research design

A qualitative study using semi-structured interviews was conducted based on the approach recommended by the National Centre for Social Research. This is based on critical realism and interpretivism using the framework approach to analysis.[31] This study is reported in line with the consolidated criteria for reporting qualitative research[32] (online supplemental file 1). The study was managed by a research steering group consisting of four researchers with subject and methods experience (HRW, CN, CMA and ACG) and two patient representatives who were former ICU patients approached through a national patient support group.

### Setting and participants

The study was based at a UK National Health Service hospital trust in London, which has four ICUs for adult patients across three hospital sites, each of which has different referring specialities. Purposive sampling[33 34] was used to recruit a range of service users (former ICU patients and their family members/carers) and the hospital's ICU clinicians from the different ICU settings, from different professional groups with a range of experience levels.

### Eligibility criteria and recruitment strategies

Clinicians were included if they were doctors (senior trainees (registrars/fellows) or consultants), nurses, occupational therapists (OTs) or physiotherapists with at least 2 months of ICU experience and who had experience of rehabilitation treatments or deciding when they should be initiated. Clinicians were approached via adverts in meetings, posted in staff areas or via general group or more targeted emails. Former ICU patients and family members were included if they had any personal experience of physical rehabilitation while in ICU. Participants were excluded if they could not attend an in-person interview, if they felt unable to participate in English, if they were less than 18 years of age or unable to give informed consent. Patients or their relatives/carers

were approached via local ICU patient support groups and follow-up clinics.

### Data saturation

Estimates were that 30 participants would be required to gain a sufficient range of perspectives, with sampling ending once apparent data saturation had been reached. Data saturation was defined *a priori* as when no new themes of barriers and facilitators were evident from interviews, as decided by the research steering group. During data collection, when the interviewer felt no new themes were being discussed, the latest version of the initial thematic framework was shown to the final clinician and patient participant as a sense check to see whether they could identify any additional themes that had been missed. Following this, the initial thematic framework was reviewed by clinical colleagues from all four professional groups included in the study, as well as the research steering group members who were former ICU patients, to discuss if any obvious themes were missing.

### Topic guide development

A semi-structured interview topic guide was developed (online supplemental file 2), designed by the research steering group (which included the input of former ICU patients) and was formatted based on typical qualitative interview procedures.[35–38] It consisted of 11 main open questions that were designed to address different aspects of the research objective (such as barriers and risk), some of which were derived from previous studies.[20 27 39] The interview was piloted with both clinicians and former patients to ensure questions were clear and fit for purpose. No modifications were required to the topic guide in response to testing.

### Interview procedures

Interviews took place at the hospital site in person and only included the interviewer and the individual participant. Each participant took part in one semistructured interview. Before the interview began, participants were asked for demographic data then the interview proceeded. The format of the interview was a conversation, where wording was not fixed and prompts were used to gain greater depth of understanding of participant views and experiences.[35] Participants were asked to define physical rehabilitation themselves; however, the study was designed based on rehabilitation consisting of mobilisation treatments ranging from exercises and movement in the bed, to mobilising out of bed and walking.[8] Participants were informed of this if they had difficulty defining rehabilitation or if their definition was markedly different from how the study conceptualised rehabilitation. Each interview was recorded and then transcribed verbatim. Transcripts were not returned to participants for review in line with current thinking about usefulness of this approach.[40]

## Reflexivity

All interviews were carried out by one interviewer (HRW) who is a male physiotherapist, working full time on the research study as part of work towards a doctorate, with training in qualitative research methods. The interviewer had previous clinical experience at several of the ICUs that were settings for this study, including working alongside some of the clinician participants, but not the patients or their relatives/carers. The researcher, therefore, had previous experiences of barriers and facilitators of rehabilitation in the study setting. These influences were taken into account using a reflexive diary before and after data collection, which was then considered during the analysis process.

## Data analysis

Thematic framework analysis[41][42] was used to produce themes based on the interview transcript data. This involves drawing up an initial list of themes that summarises all the interview data (online supplemental file 2). Data were then arranged in a framework table which structured what each participant had said about each initial theme in an easily accessible form. This facilitated the production of a final set of themes and subthemes and comparison of how these vary between groups of participants. Analysis was facilitated by the use of NVivo V.11 software (QSR International) and carried out by the first author (HRW). A second researcher (MJ) reviewed 10% of interview transcripts and confirmed that they matched the initial set of themes. At several stages during the analysis process, the research steering group met to review the data, discuss uncertainties over formation of themes and as a check on the process. Descriptive statistics were used to summarise demographic data using IBM SPSS Statistics V.25. Continuous data were tested for normality using the Shapiro-Wilk test and non-normally distributed data described using median and IQR, and normally distributed data described using mean and SD.

## Patient and public involvement

Former ICU patients were members of the research steering group. These patient representatives edited the wording of recruitment materials and inputted into the design of the topic guide. They also assisted the interviewer (HRW) to practise interview technique and were involved in reviewing the initial thematic framework, as part of data saturation checks. These patient representatives did not participate or contribute data to the study itself.

## RESULTS

Recruited participants included 16 clinicians, from a range of professions, with a range of experience in different settings (table 1). Eleven former ICU patients and three family members/caregivers participated (table 2), including substantial patient experience of ICU rehabilitation (mean patient participant length of

**Table 1** Clinician participant demographics

| | Clinicians (N=16) |
|---|---|
| Age, mean (± SD) | 34 (8.6) |
| Female, n (%) | 12 (75) |
| Profession, n (%) | |
| Doctor | 4 (25) |
| Nurse | 5 (31) |
| Therapist (physiotherapist or occupational therapist) | 7 (44) |
| Seniority, n (%) | |
| Team leader | 9 (56) |
| Senior clinician | 6 (38) |
| Junior clinician | 1 (6) |
| Number of years of ICU experience, median (IQR) | 6 (1-15) |
| Number of years of clinical healthcare experience, mean (±SD) | 11 (8) |
| Place of work*, n (%) | |
| Intensive care 1 | 5 (31) |
| Intensive care 2 | 6 (38) |
| Intensive care 3 | 5 (31) |
| Intensive care 4 | 5 (31) |
| Involvement in physical rehabilitation, n (%) | |
| Participating in the decision over whether a patient is stable enough to mobilise | 16 (100) |
| Leading rehabilitation treatment | 10 (63) |
| Assisting with rehabilitation treatment | 12 (75) |

*Some clinicians work on more than one ICU.
ICU, intensive care unit.

ICU stay: 15 days, SD±10.8 days). Initially, 53 potential participants expressed interest in taking part, of whom 30 were recruited before data saturation was achieved. Five declined or were not available for interview, three did not respond further after initial contact, data saturation was achieved before four were recruited and 11 were not recruited as others were chosen instead to gain a greater range of views, as per the purposive sampling strategy. Interviews lasted for a mean 43 min (SD±11 min).

The study themes and subthemes are described in detail below. Online supplemental material illustrates these themes and subthemes with verbatim quotes from participants, with participant numbers (see online supplemental file 2 to link quote numbers with data).

### Safety and physiological concerns

Clinician and patient participant concerns over the safety of rehabilitation were reported as a barrier to rehabilitation. This included the risk of dislodging lines and attachments, such as ventilator tubing and femoral lines (quote 1). However, some participants, who were mostly

**Table 2** Patient and caregiver participant demographics

| | Former ICU patients and caregivers (N=14) |
|---|---|
| Service user participants: | |
| Patients, n (%) | 11 (79) |
| Caregivers, n (%) | 3 (21) |
| Age, mean (±SD) | 65 (10.7) |
| Male, n (%) | 10 (71) |
| Patient ICU length of stay in days* (n=11), mean (±SD) | 15 (10.8) |
| Patient stated reason for admission (n=11), n (%) | |
| Aortic dissection | 1 (9) |
| Cardiac arrest | 1 (9) |
| Gastrointestinal | 1 (9) |
| Organ failure | 1 (9) |
| Septic shock | 1 (9) |
| Surgery | 5 (45) |
| Trauma | 1 (9) |
| Site where ICU was experienced, n (%) | |
| Intensive care 1 | 2 (14) |
| Intensive care 2 | 11 (79) |
| Intensive care 3 | 0 (0) |
| Intensive care 4 | 1 (7) |
| Highest level of physical rehabilitation experienced, n (%) | |
| Moving in bed | 2 (14) |
| Sitting in a chair | 6 (43) |
| Walking | 6 (43) |

*Two participants reported their length of stay as approximate.
ICU, intensive care unit.

clinicians, did not perceive this as a barrier, if careful planning and also organisation of the bed space environment was carried out. For example, avoiding the use of femoral vascular catheters as access for haemofiltration or planning breaks in haemofiltration could enable rehabilitation. Endotracheal tubes or airways that had been difficult to insert, were also cited as barriers, with the difficulty of titrating sedation for a balance between tube tolerance and patient alertness cited as one explanation by a clinician.

Clinician participants identified particular patient groups with barriers to rehabilitation because they felt they were at an increased risk or they presented additional logistical challenges, such as those with multiple traumatic injuries (quote 2). They suggested that patients admitted after surgery could have certain surgical precautions which presented logistical issues contacting different teams to gain clarity over safety of rehabilitation. Despite this, patients in ICU after elective surgery could have received preoperative education or preplanned rehabilitation programmes, both facilitating rehabilitation post-operatively.

Physiological instability, such as instances of respiratory distress or cardiovascular instability, was reported as preventing rehabilitation treatments by participants who were mostly clinicians.

> …it's mainly blood pressure related for me, or their resp[iratory] rate. If I don't think they're going to tolerate mobilising, and if it's going to cause more harm than good. (Therapist 2, quote 3)

Dependence on organ support, such as the amount of respiratory support or vasoactive drugs were also identified as barriers. Clinician opinion ranged from perceiving patients receiving vasoactive drugs as a contraindication to rehabilitation (quote 4), to others who considered rehabilitation possible if a low or weaning dose was used or if the patient was less severely unwell, for example, if vasoactive drugs were being used for epidural-induced hypotension. Risk relating to hypotension during rehabilitation was suggested by a clinician to relate to anxiety from junior staff about managing vasoactive drugs during mobilisation (quote 5). Some clinician participants suggested potential organ support barriers should be discussed with the ICU doctors and also advocated actively sedating patients less.

Patient participants sometimes reported feeling too unwell to actively participate in rehabilitation. Some patients reported profound feelings of weakness, making their bodies feel 'like a lead weight', which came as a surprise when they first tried to get up and was linked with feelings of vulnerability (quotes 6 and 7). These participants did then identify a time in their recovery where these symptoms subsided to the point where they could then participate.

Additionally, level of alertness, confusion and agitation, cognitive impairments and personality disorders were all cited as barriers by clinicians (quote 8). Some patients and relatives recalled experiences of delirium and hallucinations as profound influences on their recovery in general.

The difference between clinicians' perception of safety and a patient's readiness to begin rehabilitation was expressed as a barrier by some clinician participants (quote 9). Some explanations included clinician fear of the unwell patient and the risk of perceived harm which caused anxiety for some (quote 10).

> …happy to cause no harm, or kind of, and no perceived harm by not mobilising someone but actively getting up and causing harm is a, always going to be a significant anxiety for staff… (Nurse 5, quote 11)

This was linked to clinician need for control over the physiological numbers, potentially leading to a reluctance to reduce that control by moving a patient out of bed (quote 12). One doctor suggested that a paradigm shift was required to address this barrier (quote 13). Another doctor said they modified targets for acceptable changes in physiological observations (such as blood pressure), to

reassure other clinicians that mobility was still safe (quote 14).

## Patient participation and engagement

Clinician participants reported experience of patients who may be reluctant to participate in rehabilitation. When asked about this theme, patient participant responses ranged from reporting enthusiastic engagement in rehabilitation, to not wishing to mobilise out of bed. Reasons cited for their reluctance included not wanting to do something perceived as potentially worsening their condition (quote 15). Furthermore, a lack of incentive or motivation to engage was discussed, as well as a feeling of weakness, which some found difficult to accept.

> …there were times when I simply didn't want to do it… Depression, … lack of energy, lack of spirits really … (Patient 7, quote 16)

Suboptimal communication between patients and clinicians was felt to be a barrier to rehabilitation by some patient and clinician participants. Suggested reasons included the little time spent by clinicians discussing rehabilitation, difficulty communicating rehabilitation goals and some sometimes showed a lack of empathy. Suggested ways of overcoming these issues included maximising a patient's ability to communicate, giving more reassurance, building up trust, showing kindness and helping patients to feel safe (quote 17). Patients valued humour from staff and felt rapport was aided by staff continuity. Patients and relatives recommended that when a patient was reluctant to mobilise, an encouraging and diplomatic approach should be balanced with assertiveness from clinicians to 'push' patients (quote 18).

Some patient participants recommended that strategies to improve patient engagement in rehabilitation should always be patient-specific. Other suggestions, mostly from clinicians, included promoting sleep at night, involving patients in planning a rehabilitation timetable, goal setting and using outcome measures to demonstrate progress (quote 19). Furthermore, education for patients and relatives at the appropriate time, around the importance of rehabilitation was suggested.

Further facilitators suggested by patients and clinicians included the use of meaningful activities and identifying key patient motivators. The importance of tailoring rehabilitation to include activities meaningful to patients (such as functional tasks and personal care activities based on previous interests) were identified to facilitate engagement within a context more readily understood by patients.

> Looking at therapy in a slightly different way and finding an activity that's meaningful to [patients], whether that's personal care or leisure activities, and through that encouraging them to… engage in that activity and then helping them to see the therapeutic value of that. (Therapist 4, quote 20)

Recognising key patient motivators such as gaining independence and dignity by being able to do more for themselves was also suggested (quote 21). Patients reported being motivated through their improvement during rehabilitation sessions, almost as a proxy for improvement from critical illness. Patient qualities of resilience, determination and a positive mental attitude were reported as a facilitator by patients themselves.

The role of family was discussed as both a barrier and facilitator. Instances were reported by some clinician participants where relatives could be reluctant for patient participation in rehabilitation. When this was discussed with patient and relative participants, responses ranged from an understanding of why this happens, to a strong disbelief that this could be the case. The role of family in encouraging patients was discussed, with some highlighting how they were motivated to improve mobility to help their family member feel better (quote 22).

## Clinician experience and knowledge

Clinician participants discussed the experience and knowledge of those carrying out rehabilitation. A lack of experience, confidence and senior support were cited as barriers (quote 23). However, some therapists also proposed those clinicians with more experience could pose a barrier. They suggested some more experienced nurses may perceive rehabilitation as outside of their role or may have spent more time in an environment where rehabilitation was not a priority. Opinions over experience as a facilitator also varied. Some emphasised that a team with the right skill mix (including adequate senior support) was important, with a nurse suggesting having more confident staff freed up time for rehabilitation. However, some therapists reported that more inexperienced nurses could be a facilitator as they have received recent training in rehabilitation. One therapist cited enthusiasm as being more important than experience to facilitate rehabilitation.

A lack of training and knowledge, including about the importance of rehabilitation, organisation and planning of sessions and therapeutic manual handling were suggested as important factors by clinicians.

> It doesn't happen because… we are not aware enough yet how important it is, or how much difference it could make, so it's not embedded in our thinking and in our behaviour… (Doctor 4, quote 24)

A popular strategy suggested by clinicians to address these barriers was through education and training for the ICU interdisciplinary team, such as through study days and experiential learning (quote 25). Additionally, the use of a rehabilitation policy and guidelines to drive implementation and aid less experienced clinicians know when to begin rehabilitation was discussed.

## Teamwork

Discussion of teamwork covered team culture, clinician roles, rehabilitation definitions and logistics. A lack of a

rehabilitation culture leading to some staff having a less proactive attitude to rehabilitation delivery was discussed.

> But a lot of it's just to do with the attitude of the individual staff member, how proactive they are and how much they believe in mobilisation as a kind of key thing (Nurse 5, quote 26)

One explanatory factor was a lack of medical leadership. Participants (mostly clinicians) suggested promoting a culture where an interdisciplinary team works together to promote rehabilitation as routine and important would facilitate implementation. A less hierarchical culture would encourage proactive team planning and problem solving, with medical leadership again emphasised as key (quote 27).

Another key barrier to rehabilitation discussed by clinicians, was differences in opinion between professions over roles and responsibilities (quote 28). Some reported that rehabilitation was perceived as only a therapist's job (quote 29). Therapists reported that there could be a lack of understanding of their role or their other responsibilities, for example, covering other clinical areas in addition to the ICU. To overcome this, clinicians suggested promoting teamwork where separate responsibilities were acknowledged and there was a willingness to crossover professional roles, with therapists empowering nurses to facilitate rehabilitation (quote 30).

Differences in opinions over roles and responsibilities were impacted on by variation in how rehabilitation was defined and delivered. This in itself may explain some of the difficulty in promoting a proactive rehabilitation culture. Clinicians sometimes limited their definition of rehabilitation to a patient sitting out in a chair (quote 31). Conversely, OT participants widened the concept of rehabilitation to encompass a 24-hour interdisciplinary approach utilising functional tasks.

> …rehabilitation is not, you know, 20 minutes with the physio or the OT every day. Really good rehabilitation is a 24 hour approach, and that – part of that is positioning a patient in bed. Part of that is ensuring the patient gets the right nutrition as well as looking at the actual physical things that they're doing. (Therapist 4, quote 32)

This may increase patient engagement and interdisciplinary involvement, by helping staff to incorporate more rehabilitation activities during the course of their normal duties, for example, during personal care activities (quote 33).

Finally, lack of staff and logistical difficulties in implementing rehabilitation were suggested as barriers by clinicians and patients. Greater investment in staffing and utilisation of healthcare support workers was suggested to address this. Logistical concerns covered the number of staff required and the duration of a rehabilitation session in competition with other unit procedures. Logistical barriers also concerned a difficulty in timing around nurses' rest breaks and staffing ratios (quote 34). Within the study ICUs, once a patient's illness severity decreased to a certain level, the nursing staffing ratio fell from one nurse to one patient to one nurse to two patients, coinciding with a potential increase in readiness for rehabilitation. Potential strategies to address these concerns include proactive planning of sessions, for example during morning team briefings. Additionally, a change to working patterns to build in more time for rehabilitation to occur was suggested.

### Equipment and environment

A lack of working specialist rehabilitation equipment was highlighted as a barrier by clinician participants (quote 35). Clinicians advocated greater investment and suggested the whole team take ownership of ensuring equipment was fixed or find funding sources for equipment replacement. Environmental concerns raised by patients and clinicians first covered practical limitations such as space to move rehabilitation equipment around the bedspace. Furthermore, a patient highlighted the nature of the ICU environment itself did not encourage them to move out of bed (quote 36).

> …you can see some bright lights and monitors, you can hear monitors going off, but you don't have the, "Crash, bang, wallops!" that you get in a general ward… but it's a capsule and a bubble, it's a weird feeling… "People think it's like being in a spaceship" and I thought, "That's such a good description" and that's how it did feel. (Patient 8, quote 37)

### Risks and benefits of rehabilitation in intensive care

Opinions over risks and benefits were explored, which closely related to safety, knowledge and attitude towards rehabilitation. Clinician ideas about risks resembled the safety issues from theme one, however, this did not necessarily mean a reluctance to mobilise (quotes 38 and 39). Most patients and relatives reported they had not worried about the risks of mobilising while in an ICU (quote 40), although some had experienced things such as dizziness and one reported passing out. Considering benefits reported by clinicians and patients, physical benefits of rehabilitation focused on the acute impact of improving physical function, including in preparation for recovery on the wards (quote 41). Suggested psychological benefits for the patient included helping mood and well-being and restoring a sense of dignity.

> …the important thing is you sense that you're not just lying there waiting to die. …so you are… you are… coming back to being a human being that wants to live. (Patient 7, quote 42)

Finally, several clinicians reported how a benefit of patient rehabilitation was the encouragement and sense of achievement it provided for staff.

The overarching theme for how to overcome barriers to physical ICU rehabilitation related to moving from a multidisciplinary approach where different professions

work together but have separate responsibilities; towards a patient-centred, interdisciplinary team approach. This was where all parties have a shared aim of providing physical rehabilitation (quotes 27 and 30). This can facilitate clinicians working together to develop a shared understanding of the definition of rehabilitation, so patients can participate in activities that are more meaningful. Furthermore, an agreement can be developed among the team, about the benefits and risks, the optimum way to deliver rehabilitation and when it is safe to start. This can then help different professions to collaborate to help to overcome barriers related to team working and to improve the ICU environment.

## DISCUSSION

This study has provided an in-depth exploration of the views of multiprofessional ICU clinicians and was strengthened by including former ICU patients and their relatives, adding to the knowledge of overcoming barriers to ICU physical rehabilitation. Primarily, this is suggested to be through a change in approach to team working, from a multidisciplinary to an interdisciplinary and patient-centred approach. This means moving from a multidisciplinary way of working where a team is made up of different professions working on their distinct priorities,[43–45] to an interdisciplinary approach where a team of different professions work together with ICU rehabilitation a priority for all. This, therefore, emphasises a shift from rehabilitation primarily being the focus of therapy staff, to one where all team members have joint accountability and identify this as a key aspect to their work, contributing in overlapping ways but also in ways relevant to their professional skills and knowledge.[46 47] This change in perspective could facilitate a change in opinion over the definition and delivery of rehabilitation towards an interdisciplinary, 24-hour approach that includes activities meaningful to patients to facilitate engagement. An interdisciplinary working model has previously been used to facilitate more efficient and effective care during critical illness and in general rehabilitation delivery. Reported outcomes have included more coordinated interprofessional working and enhanced delivery of appropriate patient care.[44 48 49]

It is interesting to see that several of the themes of barriers and facilitators to ICU rehabilitation are similar to previous qualitative studies. This includes themes of safety,[23 24 27 28 50] patient engagement,[29] knowledge and experience[21–26] and team work.[20 22–25 27–29 50] Patient reports of experiencing feelings of weakness and vulnerability in this present study have also previously been identified[30 51] where their vulnerability may be explained, at least in part, by patients adjusting to being critically ill while having little or no memory of their deterioration into critical illness.[52] In our study, clinicians expressed differences in opinion over roles and responsibilities towards rehabilitation as well as safety concerns for initiating treatment. Staff confidence in rehabilitation

provision may contribute towards differences in viewpoints and engagement, particularly towards opinions on readiness of a patient to begin rehabilitation. This may be partially explained through differences in personality traits, between those more or less able to make pragmatic concessions to adjust to the limitations of the working environment to ensure reasonable care is delivered and to tolerate greater variability in acceptable target physiological observations.[53] This would, therefore, represent an important factor to address with staff when overcoming barriers to rehabilitation, for example, to achieve the paradigm shift suggested by some participants to enable clinicians to address anxiety in relation to control over physiological parameters.

This study has added to previous knowledge in several ways. Interestingly and perhaps surprisingly, less clinical experience was highlighted as a potential facilitator of rehabilitation. Some therapists reported that more inexperienced nurses have received recent training in rehabilitation and one therapist cited enthusiasm as more important than experience. Additionally, perceptions of the content of rehabilitation were notable. Some viewed rehabilitation as being limited to sitting in a chair. This can contribute to a limited scope of rehabilitation practice[26] and may have contributed to the lack of rehabilitation culture reported by some participants in this study. The OTs emphasised the inclusion of personal care activities as part of ICU rehabilitation delivered at any point in the day by any profession, to facilitate a more positive rehabilitation culture. This is supported by Laerkner et al[54] who compared the views of nurses and patients in Denmark, and found nurses recommended incorporating familiar activities into rehabilitation and patients emphasised the importance of empathy and compromise from clinicians. While patients in this present study agreed with Laerkner's recommendations of clinician–patient communication, they also emphasised that at times, a more assertive approach from clinicians in encouraging rehabilitation is desirable.

The findings from this study focus us to create a patient-centred interdisciplinary approach to rehabilitation. This involves considering how clinicians communicate with patients and broadening the definition of rehabilitation to include functional tasks that are meaningful to patients. Furthermore, broadening delivery of rehabilitation to a 24-hour holistic approach that includes family members,[55–57] with a focus on prioritising patient-reported motivators of independence and dignity and to progress back towards normality. Facilitating this change in a multifaceted ICU environment would benefit from using implementation and improvement science methodology, where codesign by ICU clinicians from different professions, as well as service users can be employed. This change in practice should be evaluated not only in terms of whether it improves rehabilitation delivery without impacting patient safety, but also in terms of how these changes influence other ICU procedures and working practices.[58]

Limitations of this study include the potential for poor recall from patient or relative/carer participants as the time from ICU admission to interview was not recorded.[52] [59] However, as participants were usually recruited at their first ICU follow-up appointment, this was unlikely to be an extended time. Furthermore, differences in use of language of some participants sometimes made it difficult to discern the exact point they were making during analysis, therefore, although this demonstrates diversity within the sample, some finer detail may have been lost. The method of approach may have meant that more patients actively engaged in the issues being evaluated were recruited. Those patients not attending follow-up appointments may have had different opinions. Pragmatic restrictions meant few family members were recruited and more patients who had experienced one of the ICU sites were involved. Finally, the application of these findings to other areas should consider that participants were included from sites in one city.

In conclusion, this exploration of a range of clinician and patient perspectives suggested a patient-centred, interdisciplinary approach to implementing ICU physical rehabilitation. These findings constitute a starting point for optimising rehabilitation delivery through improvement and implementation science.

**Acknowledgements** We thank staff at the research site for their support in facilitating recruitment and data collection. We thank Professor Alison McGregor (Imperial College London) for advice on developing the overarching theme. Data from this study have previously been presented at the Intensive Care Society State of the Art meeting 2018 and 2021, the 6th European Conference on Weaning and Rehabilitation in Critically Ill Patients 2018, Physiotherapy UK conference 2018 and the North West London Research Symposium for Health Professions 2018.

**Contributors** HRW, SJB, CMA and ACG contributed to the conception and planning of this work. HRW, CN, MJ, SJB, CMA and ACG contributed substantially to the design of this work. HRW carried out recruitment of participants with assistance from SJB. HRW also completed data collection and CN, MJ, SJB, CMA and ACG advised on the conduct of the study. HRW had main responsibility for carrying out analysis; MJ, CN and CMA assisted in checking development of themes and HRW, CN, MJ, SJB, CMA and ACG advised on interpretation of data. HRW drafted this report, and all authors revised it critically for important content and approved the final published version. All authors are accountable for the accuracy and integrity of the work. HRW is the guarantor.

**Funding** This report is independent research arising from a Clinical Doctoral Research Fellowship, awarded to HRW, (ICA-CDRF-2015-01-026), supported by the National Institute for Health Research (NIHR) and Health Education England. We acknowledge the support of the NIHR Clinical Research Network and infrastructure support for this research was provided by the NIHR Imperial Biomedical Research Centre (BRC). HRW is supported by the NIHR Imperial BRC.

**Disclaimer** The views expressed in this publication are those of the authors and not necessarily those of the NHS, the NIHR, Health Education England or the Department of Health.

**Competing interests** HRW reports personal lecture fees from Brunel University and the University of East London. CN reports personal lecture fees from Janssen Pharmaceuticals and Web MD. CMA is a National Institute of Health Research PhD fellowship committee member, although not at the time that funding was awarded for this study. ACG reports salary support from an NIHR Research Professorship (RP-2015-06-018) and research support from the NIHR Imperial BRC.

**Patient and public involvement** Patients and/or the public were involved in the design, or conduct, or reporting, or dissemination plans of this research. Refer to the Methods section for further details.

**Patient consent for publication** Not applicable.

**Ethics approval** This study involves human participants and was approved by London – Bloomsbury Research Ethics Committee (17/LO/0362). Participants gave informed consent to participate in the study before taking part.

**Provenance and peer review** Not commissioned; externally peer reviewed.

**Data availability statement** No data are available beyond what is reported in this manuscript and the supplemental files due to participant confidentiality requirements.

**ORCID iDs**
Huw R Woodbridge http://orcid.org/0000-0003-3823-9757
Stephen J Brett http://orcid.org/0000-0003-4545-8413

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
