## [Reviewer comments · BMJ Open]

ARTICLE DETAILS

TITLE (PROVISIONAL)	Clinician and patient perspectives on the barriers and facilitators to physical rehabilitation in intensive care: a qualitative interview study
AUTHORS	Woodbridge, Huw; Norton, Christine; Jones, Mandy; Brett, Stephen; Alexander, Caroline; Gordon, Anthony

VERSION 1 – REVIEW

REVIEWER	Nickels, Marc Princess Alexandra Hospital, Physiotherapy Department
REVIEW RETURNED	01-May-2023

GENERAL COMMENTS	Overview: The authors present a qualitative study using a thematic framework analysis to describe patient, relative/carer, and clinician perceptions of barriers and facilitators to early physical rehabilitation in the ICU. Following purposeful sampling from three London hospital sites (that included 4 ICUs) 11 former patients and 16 clinicians completed semi-structured interviews. Only 3 family members were interviewed. The primary facilitator was suggested to be changing from a multi-disciplinary to an interdisciplinary model of care were all clinicians have a shared aim of providing physical rehabilitation. Introduction: Consider including other point prevalence studies to give international perspective regarding low levels of delivery of early rehabilitation (1-4) . Methods: Typo page 6, line 18. Missing 'l' in 'whilst' To be eligible to be included as a patient participant, how much experience did the of early rehabilitation did the patient participants need to have had?
--

	Further description of the study steering group would be beneficial. How many people did the study steering group consist of? How were the members of the study steering group chosen? Did members of the study steering group provide also patient perspectives in this study? Is the study steering group the same as the research steering group? If so, I recommend consistency in terminology. If not, recommend describing the membership of the different steering groups. In the sentence: (Page 6 lines 38-44). 'Following this, the initial thematic framework was reviewed by clinical colleagues from all four professional groups included in the study, as well as steering group members who were former ICU patients, to discuss if any obvious themes were missing.' Which steering group are the authors referring to here? Are all members of the steering group described former ICU patients? If not, add 'the' prior to the 'insert research or study' steering group members who were former ICU patients,' If so, describe the composition of the appropriate steering group as per earlier comments. Regarding the interview, it is implied that the interviews are in person. Suggest making the format of the interviews explicit by adding 'Face to face' or 'in person' prior to the sentence at the commencement of the paragraph (page 6 line 46). Following piloting of the interview were modifications made to the interview? If so, please note this. (The specific modifications do not need to be described). Please note the version number of the IBM SPSS software. Page 8 Patient and public involvement. It appears implied that research steering group members provided input into the development and checking of the data. It would be worth noting explicitly whether research steering group members responses were included as analysed data. Results: Inconsistency noted in describing participants. It is unclear if participants are clinicians or patients or both, when not explicitly stated. For example, page 11. Lines 46, 50, page 12. Line 47, page 13. Line 25, page 14 line 5. Discussion I would like some discussion around the concepts reported in 1.5 with regards to differences in clinicians' opinions regarding readiness to begin mobilisation. For example, Quote 11 is rich with
--	---

information regarding the perception of not causing harm by not commencing early rehabilitation, whilst simultaneously implying that they are aware that prolonged bed rest is also a cause of harm. Subsequent quotes discuss the requirements for a paradigm shift in mind-set to tolerate a certain amount of physiological variability.

Limitations should acknowledge that the sample was taken from a relatively small geographical area and hence the generalisability of findings to other settings is likely to be limited.

Reference for including family members would be beneficial Page 19, line 53. Examples below:

1. Marshall AP, Lemieux M, Dhaliwal R, Seyler H, MacEachern KN, Heyland DK. Novel, Family-Centered Intervention to Improve Nutrition in Patients Recovering From Critical Illness: A Feasibility Study. *Nutr Clin Pract.* 2017;32(3):392-9.
2. Marshall AP, Wake E, Weisbrodt L, Dhaliwal R, Spencer A, Heyland DK. A multi-faceted, family-centred nutrition intervention to optimise nutrition intake of critically ill patients: The OPTICS feasibility study. *Aust Crit Care.* 2016;29(2):68-76.
3. Heyland DK, Davidson J, Skrobik Y, des Ordon AR, Van Scoy LJ, Day AG, Vandall-Walker V, Marshall AP. Improving partnerships with family members of ICU patients: study protocol for a randomized controlled trial. *Trials.* 2018;19(1):3.

The limited recall of exercise interventions in ICU has been noted before and could be referenced to substantiate this phenomenon. (5)

References:

1. Berney SC, Harrold M, Webb SA, Seppelt I, Patman S, Thomas PJ, Denehy L. Intensive care unit mobility practices in Australia and New Zealand: a point prevalence study. *Crit Care Resusc.* 2013;15(4):260-5.
2. Jolley SE, Moss M, Needham DM, Caldwell E, Morris PE, Miller RR, Ringwood N, Anders M, Koo KK, Gundel SE, Parry SM, Hough CL, Acute Respiratory Distress Syndrome Network I. Point Prevalence Study of Mobilization Practices for Acute Respiratory Failure Patients in the United States. *Crit Care Med.* 2017;45(2):205-15.
3. Sibilla A, Nydahl P, Greco N, Mungo G, Ott N, Unger I, Rezek S, Gemperle S, Needham DM, Kudchadkar SR. Mobilization of Mechanically Ventilated Patients in Switzerland. *J Intensive Care Med.* 2020;35(1):55-62.
4. Timenetsky KT, Neto AS, Assuncao MSC, Taniguchi L, Eid RAC, Correa TD, e Mg. Mobilization practices in the ICU: A nationwide 1-day point- prevalence study in Brazil. *PLoS One.* 2020;15(4):e0230971.
5. Nickels MR, Aitken LM, Barnett AG, Walsham J, McPhail SM. Acceptability, safety, and feasibility of in-bed cycling with critically ill patients. *Aust Crit Care.* 2020;33(3):236-43.

REVIEWER	Gaid, Dina McGill University
REVIEW RETURNED	20-May-2023

GENERAL COMMENTS	Thank you for inviting me to review this valuable work. This paper provides an in-depth exploration of the views of multi-professional ICU clinicians. However, I have a number of suggestions to strengthen the paper that needs to be considered by the authors. Abstract: The results section reads “The overarching theme related to how barriers can be overcome by moving away from a multidisciplinary approach and towards an interdisciplinary, patient-centred approach to ICU physical rehabilitation” This part needs more clarity. Methods: Methods sections need to be more organized. I suggest adding sub-titles such as research design, participants and setting, eligibility criteria, recruitment strategies, procedures, and data analysis. Also, you need to add “Data collection tool” to describe how you develop the interview guide, based on what framework (previous studies)? How did you test for content and face validity before using this guide? What domains are covered? How many questions are included? Did it last for how long? Results You listed barriers and facilitators related to each theme together, while I would suggest separating them into “facilitators” and “barriers” with strategies to overcome them. This strategy will make the results section easier for readers to track. The way you currently list them in the results section is different than how you present them in the initial thematic framework (in the appendix).
--

VERSION 1 – AUTHOR RESPONSE

Reviewer: 1

Introduction:

Consider including other point prevalence studies to give international perspective regarding low levels of delivery of early rehabilitation (1-4) .

We have added the suggested references to the last sentence of the first paragraph on page 4: citations 14-17 in the reference list.

Methods:

Typo page 6, line 18. Missing 'l' in 'whilst'

Typo has been corrected.

To be eligible to be included as a patient participant, how much experience did the of early rehabilitation did the patient participants need to have had?

We did not put any limits on the amount of experience of rehabilitation by patient participants. To make this clear, we have added “any” personal experience to the inclusion criteria:

“Former ICU patients and family members were included if they had any personal experience of physical rehabilitation whilst in ICU.” [Eligibility criteria and recruitment strategies, methods section, page 5]

However, largely because of how we recruited, we ended up with a patient cohort with substantial ICU experience. We have therefore highlighted this in the results section:

“Eleven former ICU patients and three family members/caregivers participated (Table 2), with substantial patient experience of ICU rehabilitation (mean patient participant length of ICU stay: 15 days, standard deviation \pm 10.8 days).” [First paragraph, results section, page 8]

Further description of the study steering group would be beneficial. How many people did the study steering group consist of? How were the members of the study steering group chosen?

Details of the research steering group have now been added to the first paragraph of the methods:

“The study was managed by a research steering group consisting of four researchers with subject and methods experience (HRW, CN, CMA, ACG) and two patient representatives who were former ICU patients approached through a national patient support group.” [Page 5]

Did members of the study steering group provide also patient perspectives in this study?

Patient representatives on the study steering group did not participate in the study or contribute data. We have now made this explicit in the patient and public involvement section:

“These patient representatives did not participate or contribute data to the study itself.” [Page 8]

Is the study steering group the same as the research steering group?

If so, I recommend consistency in terminology. If not, recommend describing the membership of the different steering groups.

They are the same group. Terminology has been amended so the group is consistently referred to as the ‘research steering group’:

- “The study was managed by a research steering group consisting of four researchers with subject and methods experience (HRW, CN, CMA, ACG) and two patient representatives who were former ICU patients approached through a national patient support group.” [Research design section, page 5]
- “Data saturation was defined a priori as when no new themes of barriers and facilitators were evident from interviews, as decided by the research steering group.” [Data saturation section, page 6]
- “Following this, the initial thematic framework was reviewed by clinical colleagues from all four professional groups included in the study, as well as the research steering group members who were former ICU patients, to discuss if any obvious themes were missing.” [Data saturation section, page 6]
- “A semi-structured interview topic guide was developed (supplemental file 2), designed by the research steering group (which included the input of former ICU patients) and was formatted based on typical qualitative interview procedures.” [Topic guide development section, page 6]

- “At several stages during the analysis process, the research steering group met to review the data, discuss uncertainties over formation of themes and as a check on the process.” [Data analysis section, page 7]
- “Former ICU patients were members of the research steering group.” [Patient and public involvement section, page 8]

In the sentence: (Page 6 lines 38-44). ‘Following this, the initial thematic framework was reviewed by clinical colleagues from all four professional groups included in the study, as well as steering group members who were former ICU patients, to discuss if any obvious themes were missing.’ Which steering group are the authors referring to here?

This is the research steering group – sentence amended to clarify this.

Are all members of the steering group described former ICU patients? If not, add ‘the’ prior to the ‘insert research or study’ steering group members who were former ICU patients,’ If so, describe the composition of the appropriate steering group as per earlier comments.

The research steering group consisted of researchers and former ICU patients. Therefore, ‘the’ has been added to the sentence to clarify this:

“Following this, the initial thematic framework was reviewed by clinical colleagues from all four professional groups included in the study, as well as the research steering group members who were former ICU patients, to discuss if any obvious themes were missing.” [Data saturation section, page 6] See response to comment above for composition of steering group members.

Regarding the interview, it is implied that the interviews are in person. Suggest making the format of the interviews explicit by adding ‘Face to face’ or ‘in person’ prior to the sentence at the commencement of the paragraph (page 6 line 46).

‘in person’ added:

“Interviews took place at the hospital site in person and only included the interviewer and the individual participant.” [Interview procedures section, page 6]

Following piloting of the interview were modifications made to the interview? If so, please note this. (The specific modifications do not need to be described).

No specific modifications were required to the interview topic guide, this has been made clear in the text:

“No modifications were required to the topic guide in response to testing.” [Topic guide development section, page 6]

Please note the version number of the IBM SPSS software.

Version 25 has been added:

“Descriptive statistics were used to summarise demographic data using IBM SPSS Statistics 25.” [Analysis section, page 7]

Page 8 Patient and public involvement.

It appears implied that research steering group members provided input into the development and checking of the data. It would be worth noting explicitly whether research steering group members responses were included as analysed data.

A sentence has been added to clarify that they did not participate or contribute any analysed data: "These patient representatives did not participate or contribute data to the study itself." [Patient and public involvement section, page 8]

Results:

Inconsistency noted in describing participants. It is unclear if participants are clinicians or patients or both, when not explicitly stated. For example, page 11. Lines 46, 50, page 12. Line 47, page 13. Line 25, page 14 line 5.

We have amended the results section to provide greater clarity on whether the perspectives of patients and/or clinicians are being discussed, at the points highlighted by the reviewer as required and at several additional places in pages 10-18.

Discussion

I would like some discussion around the concepts reported in 1.5 with regards to differences in clinicians' opinions regarding readiness to begin mobilisation. For example, Quote 11 is rich with information regarding the perception of not causing harm by not commencing early rehabilitation, whilst simultaneously implying that they are aware that prolonged bed rest is also a cause of harm. Subsequent quotes discuss the requirements for a paradigm shift in mind-set to tolerate a certain amount of physiological variability.

We agree that this is certainly one of the most interesting findings from our study, as well as being of practical use to clinicians in facilitating a rehabilitation culture on ICU. We attempted to take this up in the second paragraph of the discussion by linking it to staff anxiety because of a lack of confidence and previous evidence of the impact of perfectionist personality traits on ICU teamworking. Staff with perfectionist personality traits may find it more challenging to seemingly compromise control over stable observations and accept more variability in the light of multiple priorities for the patient. On reflection, this could have been clearer that we were referring to this part of the data. We have therefore amended this paragraph to make this more explicit:

"Staff confidence in rehabilitation provision may contribute towards differences in viewpoints and engagement, particularly towards opinions on readiness of a patient to begin rehabilitation. This may be partially explained through differences in personality traits between those more or less able to make pragmatic concessions to adjust to the limitations of the working environment to ensure reasonable care is delivered and to tolerate greater variability in acceptable target physiological observations⁵³. This would therefore represent an important factor to address with staff when overcoming barriers to rehabilitation, for example, to achieve the paradigm shift suggested by some participants to enable clinicians to address anxiety in relation to control over physiological parameters." [Discussion, page 19]

Limitations should acknowledge that the sample was taken from a relatively small geographical area and hence the generalisability of findings to other settings is likely to be limited.

This limitation has been added to the end of the limitations paragraph:

“Finally, the application of these findings to other areas should consider that participants were included from sites in one city.” [Discussion, page 20]

Reference for including family members would be beneficial Page 19, line 53.

References for including family members in rehabilitation provision have been added (references 55-57):

55. Heyland DK, Davidson J, Skrobik Y, et al. Improving partnerships with family members of ICU patients: study protocol for a randomized controlled trial. *Trials* 2018;19:3.

56. Marshall AP, Wake E, Weisbrodt L, et al. A multi-faceted, family-centred nutrition intervention to optimise nutrition intake of critically ill patients: The OPTICS feasibility study. *Australian Critical Care* 2016;29:68-76.

57. Haines KJ. Engaging Families in Rehabilitation of People Who Are Critically Ill: An Underutilized Resource. *Phys Ther* 2018;98:737-44.

The limited recall of exercise interventions in ICU has been noted before and could be referenced to substantiate this phenomenon. (5)

The reference suggested by the reviewer, plus one by Corner et al. (2019) who also recorded this phenomenon has now been added to the discussion (references 52 and 59):

52. Corner EJ, Murray EJ, Brett SJ. Qualitative, grounded theory exploration of patients' experience of early mobilisation, rehabilitation and recovery after critical illness. *BMJ Open* 2019;9:e026348.

59. Nickels MR, Aitken LM, Barnett AG, et al. Acceptability, safety, and feasibility of in-bed cycling with critically ill patients. *Aust Crit Care* 2020;33:236-43.

Reviewer: 2

Abstract:

The results section reads “The overarching theme related to how barriers can be overcome by moving away from a multidisciplinary approach and towards an interdisciplinary, patient-centred approach to ICU physical rehabilitation” This part needs more clarity.

This sentence has been amended to clarify the meaning of multidisciplinary and interdisciplinary approaches and to explain the overarching theme in clearer language:

“The overarching theme for overcoming barriers was a change in working model from ICU clinicians having separate responsibilities (a multidisciplinary approach) to one where all parties have a shared aim of providing patient-centred ICU physical rehabilitation (an interdisciplinary approach).” [Abstract, page 2-3]

Methods:

Methods sections need to be more organized.

I suggest adding sub-titles such as research design, participants and setting, eligibility criteria, recruitment strategies, procedures, and data analysis.

Many thanks for recommending this formatting change. Relevant sub-headings have now been added and we agree this makes the methods much more clearly structured. The methods section therefore now has the following sub-titles:

- Research design
- Setting and participants
- Eligibility criteria and recruitment strategies
- Data saturation
- Topic guide development
- Interview procedures
- Reflexivity
- Data analysis
- Patient and public involvement

Also, you need to add “Data collection tool” to describe how you develop the interview guide, based on what framework (previous studies)? How did you test for content and face validity before using this guide? What domains are covered? How many questions are included? Did it last for how long?

An extra paragraph has been added to the methods giving greater detail on the development of the interview topic guide:

“Topic guide development

A semi-structured interview topic guide was developed (supplemental file 2), designed by the research steering group (which included the input of former ICU patients) and was formatted based on typical qualitative interview procedures 35-38. It consisted of 11 main open questions that were designed to address different aspects of the research objective (such as barriers and risk), some of which were derived from previous studies^{20 27 39}. The interview was piloted with both clinicians and former patients to ensure questions were clear and fit for purpose. No modifications were required to the topic guide in response to testing.” [Methods, page 6]

We ask whether the reviewer would consider us using “Topic guide development” as the sub-title to directly refer to the tool we were using for these qualitative methods. We also request that the length of the interviews could remain in the first paragraph of the results.

Results

You listed barriers and facilitators related to each theme together, while I would suggest separating them into “facilitators” and “barriers” with strategies to overcome them. This strategy will make the results section easier for readers to track. The way you currently list them in the results section is different than how you present them in the initial thematic framework (in the appendix).

We originally considered setting out barriers and facilitators separately as the reviewer suggests. However, after initial drafts we concluded that this would make the results section less clear to read. Many of the barriers and facilitators are closely interlinked, therefore it felt sensible to link them together in the text. Furthermore, if they were to be separated out, this would lead to a greater amount of repetition in the text, increasing the length of the results section.

The initial thematic framework included in the appendix, represents an earlier stage in the analysis before we made the decision to combine the barriers and facilitators together. This is helpful to include in the supplementary material for transparency of the stages of the analysis process, but does not represent the final themes resulting from the analysis.

We therefore request the reviewer would allow us to keep the structure of the results section the same, so we can represent how barriers and facilitators are interlinked and to prevent repetition and increased length of paragraphs.

VERSION 2 – REVIEW

REVIEWER	Nickels, Marc Princess Alexandra Hospital, Physiotherapy Department
REVIEW RETURNED	19-Sep-2023
GENERAL COMMENTS	Thank-you for addressing the majority of the recommendations made in your revised manuscript. I believe the revised manuscript is well written and improved clarity has been achieved. No further recommendations.

VERSION 2 – AUTHOR RESPONSE